# Catalyzed Hydrothermal Pretreatment of Oat Husks for Integrated Production of Furfural and Lignocellulosic Residue

**DOI:** 10.3390/polym16050707

**Published:** 2024-03-05

**Authors:** Maris Puke, Daniela Godina, Prans Brazdausks

**Affiliations:** Latvian State Institute of Wood Chemistry, Dzerbenes 27, LV-1006 Riga, Latvia; daniela.godina@kki.lv (D.G.); prans.brazdausks@kki.lv (P.B.)

**Keywords:** catalyzed hydrothermal pretreatment, oat husks, furfural, acetic acid, lignocellulose

## Abstract

This study presents a novel approach for biorefining oat husks into furfural, leveraging a unique pilot-scale setup. Unlike conventional furfural manufacturing processes, which often result in substantial cellulose degradation and environmental concerns associated with sulfuric acid usage, our method utilizes phosphoric acid as a catalyst to achieve high furfural yield while minimizing cellulose destruction. Drawing on our research conducted in a distinctive pilot-scale environment, we successfully developed and implemented a tailored biorefining process for oat husks. Through meticulous experimentation, we attained a remarkable furfural yield of 11.84% from oven-dried mass, accompanied by a 2.64% yield of acetic acid. Importantly, our approach significantly mitigated cellulose degradation, preserving 88.31% of the cellulose content in oat husks. Existing catalytic (H_2_SO_4_) furfural manufacturing processes often lead to substantial cellulose degradation (40–50%) in lignocellulosic leftover during the pretreatment stage. As a result of the research, it was also possible to reduce the destruction of cellulose in the lignocellulose leftover to 11.69% of the output (initial) cellulose of oat husks. This research underscores the feasibility and sustainability of utilizing oat husks as a valuable feedstock for furfural production, highlighting the potential of phosphoric acid as a catalyst in biorefining processes. By showcasing our unique pilot-scale methodology, this study contributes to advancing the field of environmentally friendly biorefining technologies.

## 1. Introduction

Hydrothermal pretreatment stands out as a promising technology for the production of biofuels and high-value products from lignocellulosic feedstocks [1,2]. The hydrothermal pretreatment of oat husks for the production of furfural and acetic acid involves subjecting the biomass to high-temperature and high-pressure conditions in the presence of water and a catalyst [3,4]. This process aims to break down the complex structure of lignocellulosic materials, such as oat husks, into simpler chemical compounds that can be further processed into valuable products like furfural and acetic acid [4]. Hydrothermal pretreatment typically involves temperatures ranging from 160 to 200 °C. Elevated pressure is applied to maintain water in a liquid state at these high temperatures [5,6]. The duration of hydrothermal treatment, or retention time, is an essential parameter that influences the extent of biomass depolymerization. The optimal duration depends on specific conditions and desired product outcomes [7,8]. In some cases, catalysts may be used to enhance the efficiency of the process. Sulfuric acid is a common catalyst employed in hydrothermal pretreatment [2,9]. Hydrothermal pretreatment of oat husks can result in the deacetylation of hemicellulose, yielding acetic acid [10,11]. Furfural is often produced from the dehydration of pentose sugars (like xylose) present in the hemicellulose fraction [12]. The conditions of hydrothermal pretreatment, including temperature, pressure, catalyst concentration, and catalyst amount for each lignocellulosic biomass type, need to be optimized to maximize the yield of furfural and acetic acid [13,14]. The liquid fraction obtained after hydrothermal pretreatment, known as hydrolysate, may undergo further processing to separate and purify the desired products [15]. It is worth noting that the successful implementation of hydrothermal pretreatment for oat husks requires a balance between breaking down complex biomass components and minimizing the degradation of valuable components. Optimization studies are crucial for achieving high yields of furfural and acetic acid while maintaining the overall efficiency and economic viability of the process [4].

MacDermid et al. [16] explored the production of furfural from hemicellulose using hydrothermal pretreatment, identifying optimal conditions as a temperature of 170 °C, a 30 min reaction time, and a sulfuric acid concentration of 0.5% (*w*/*w*). These conditions yielded a furfural output of 63.5%, with increasing acid concentration and reaction time positively impacting furfural yield. The study concluded that hydrothermal pretreatment is a promising technology for furfural production from hemicellulose. In a separate investigation, Zhang et al. [10] delved into furfural production from xylose through hydrothermal pretreatment. Optimal conditions were determined as a temperature of 180 °C, a 30 min reaction time, and a sulfuric acid concentration of 0.5% (*w*/*w*), resulting in a furfural yield of 68.5%. Similar to the previous study, increasing acid concentration and reaction time was found to enhance furfural yield. The authors concluded that hydrothermal pretreatment holds promise for furfural production from xylose.

After the hydrothermal pretreatment of oat husks to obtain furfural and acetic acid, the remaining lignocellulose can still be a valuable resource that can be utilized for various purposes [16]. The residual lignocellulose can be processed to obtain nanofibrillated cellulose (NFC), which consists of nano-sized fibers. NFC has various applications, including as a reinforcing agent in composite materials and papermaking and as a rheology modifier in different products [17]. The lignocellulosic residue can also be further processed through enzymatic hydrolysis or other methods to release sugars. These sugars can then be fermented to produce biofuels such as ethanol, which is a sustainable and renewable energy source [18]. Also, the lignocellulosic residues can be incorporated into biocomposite materials, offering a sustainable alternative to traditional materials. These composites can find applications in construction, packaging, and automotive industries [19]. The remaining lignocellulose can be used as a soil amendment to improve soil structure and fertility. It adds organic matter to the soil, enhancing its water retention capacity and promoting microbial activity [20]. Because processed lignocellulose is so absorbent, it can be used as animal bedding. It can also be added to animal feed formulas to supply dietary fiber [21]. The lignocellulosic residue can be used as a raw material for the production of biodegradable products, reducing reliance on non-biodegradable materials [22]. Of course, lignocellulosic residues can be burned for thermal energy generation. The utilization of the remaining lignocellulose is highly dependent on the composition of the residual biomass, the effectiveness of the hydrothermal pretreatment, and the targeted applications. Integrating a biorefinery approach can enhance overall sustainability by maximizing the utilization of different components within the biomass [23].

The primary objective of this research is to develop an innovative biorefining technique (Figure 1) aimed at transforming oat husks into furfural and acetic acid, all the while ensuring the preservation of cellulose within the resulting lignocellulose residue [24]. 

The remaining cellulose obtained after the hydrothermal pretreatment of oat husks will potentially be used for 5-HMF (5-hydroxymethylfurfural) production. 5-HMF is a versatile platform chemical that can be derived from various biomass sources, including cellulose-rich materials [25]. It is important to note that the success of this process depends on reaction condition optimization, including temperature, pressure, and catalyst selection, to favor the formation of 5-HMF over other by-products [26]. The potential utilization of the remaining cellulose for 5-HMF production would require a comprehensive understanding of the specific composition of the cellulose-rich residue obtained from the oat husks and the development of a tailored conversion process.

## 2. Materials and Methods

### 2.1. Materials and Chemicals

Orthophosphoric acid (H_3_PO_4_) (85%), sulfuric acid (95–97%), D-(+)-cellobiose (≥99%), D-(+)-glucose, (≥99.5%), D-(+)-xylose (≥99%), L-(+)-arabinose (≥99%), D-(+)-galactose (≥99%), D-(+)-mannose (≥99%), furfural (≥98.5%), acetic acid (≥99.5%), 5-hydroxymethylfurfural (5-HMF) (≥98.5%), levulinic acid (≥98%), and formic acid (≥99%) were purchased from Merck (Eschenstr. 582024 Taufkirchen, Germany) and used without further purification.

### 2.2. Feedstock

The oats harvested in the EU were 7.5 million tons in 2022, but in Latvia, they were 90,000–100,000 ha per year, and the yield was 4–6.5 t/ha [27]. Oat husks are a significant part of the seeds—up to 25%. Oat husks were supplied by a Latvian company, A/S Dobeles Dzirnavnieks, and used directly without any mechanical modifications. The company deals with cereal processing and various products (flour products, grocery products, flakes, etc.). 

### 2.3. Characterization of Feedstock

Initially, we quantified the extractives in the oat husks, adhering to the TAPPI 204 cm-07 standard [28]. The extraction was carried out using Knöfler-Böhm extractors with acetone, and it lasted for 6 h. To isolate the structural carbohydrates (like glucose, xylose, galactose, arabinose, and mannose) from the oat husk samples, we employed a two-stage sulfuric acid hydrolysis method according to the National Renewable Energy Laboratory (NREL) standard TP-510-42618 [29]. Saccharide concentrations were then determined using HPLC on a Shimadzu LC20AD liquid chromatograph (Kyoto, Japan) equipped with a Shimadzu RID 10A RI detector and a Thermo Scientific HyperREZ XP Carbohydrates Pb^2^⁺ column (Waltham, MA, USA). The analysis used Milli-Q water as a mobile phase with a flow rate of 0.6 mL/min at an oven temperature of 70 °C. Before the HPLC assessments, barium carbonate was employed to counteract sulfuric acid effects. Moreover, we evaluated by-products of the hydrolysis process, including formic acid, acetic acid, levulinic acid, 5-hydroxymethylfurfural, and furfural, using HPLC but without prior neutralization. This particular analysis utilized a Shodex Sugar SH-1821 column at 50 °C, with 0.005 M H_2_SO_4_ as the eluent, and maintained a flow rate of 0.6 mL/min. All analytical standards were sourced from Merck (Darmstadt, Germany). Finally, acid-insoluble lignin and ash content were assessed based on the NREL TP-510-42618 and TP-510-42622 [30] standards, respectively.

### 2.4. Catalyzed Pretreatment of Oat Husks

Oat husks (moisture content Wrel. = 8.47%) were mixed with a catalyst solution in a specially designed blade-type mixer. A diluted orthophosphoric acid solution of a varied concentration was used as a catalyst. A total of 1360 g (o.d.m.) of oat husks, as well as a catalyst, whose quantity varied based on the experimental design, were used in each experiment. After mixing the oat husks with a defined amount of the catalyst, the obtained material was treated with a continuous superheated steam flow in an original bench-scale reactor system (shown in Figure 2) that allows for the modulation of the industrial furfural production process. The diameter of the main reactor camera was 110 mm, its height was 1450 mm, and it had a volume of 13.7 L and a max pressure of 1.2 MPa. The reactor has two heat insulation systems with automatic equipment to ensure a constant temperature in the reaction zone during the whole process time and with different process parameters. 

After the treatment process, the obtained hydrolysate contained value-added products such as formic acid, acetic acid, levulinic acid, 5-HMF and furfural. The amounts of these products were analyzed with HPLC using a Shimadzu LC20AD liquid chromatograph equipped with an RI detector (Shimadzu RID 10A) and a Shodex Sugar SH-1821 column (method described in Section 2.2).

The treated oat husk (lignocellulosic residue (LCR)) was discharged from the reactor and weighed, and the moisture content was determined by the automatic infrared moisture analyzer Shimadzu MOC–120H according to the NREL/TP-510-42621 standard [31]. Before chemical analysis, the oat husk LCR was dried to the moisture content of 10% in a forced air circulation chamber Venticell 707 ECO line at 55 °C for 24 h and grounded in a Retsch GmbH SM100 cutting mill (used sieve 0.75 mm) (Haan, Germany). Carbohydrates, acetyl groups, and acid-soluble and acid-insoluble lignin in the oat husk LCR were analyzed according to the NREL TP-510-42618 standard. HPLC procedures are described in Section 2.2. All yields of the products were calculated on the oven-dried mass of the initial feedstock. For each sample, two parallel experiments were carried out, and the obtained results are shown as the average, with the relative standard deviation (RSD) for all experiments being less than 5%.

### 2.5. Experimental Design

Previous studies [32,33,34,35,36,37,38] showed that there are a lot of technological parameters that can affect the outcome of target biobased products. Therefore, the effect of the variables of catalyst concentration (c), temperature (T), catalyst amount (m), treatment time (τ) has been studied (see Table 2). In turn, the constant factors were the moisture of the raw material (w)—8.47%—and the steam flow rate (v)—100 mL·min^−1^. The response surface methodology (an efficient tool for establishing the relationship between the interesting variables and the obtained responses) using the central composite circumscribed (CCC) design method was used to investigate the effect of experimental variables. The central point of the experimental plan can be seen in Table 2. Run 3. Data analysis was performed using Design-Expert 13 software (Stat-Ease Inc., Minneapolis, MN, USA). After their implementation, it was possible to judge the direction in which to continue the experimental work.

Experimental work was performed on an original bench-scale reactor system (described in Section 2.4). A total of 26 experimental trials were performed. Two of them were center-point replicates.

## 3. Results and Discussion

### 3.1. Analysis of the Raw Material

As known, the price of feedstock is one of the crucial factors in producing a single product. These costs are highly variable, depending on plant size, location, and used technology. To reduce the impact of the feedstock price on the final product, it is necessary to produce more than one bioproduct on the spot. To evaluate what bioproducts can be obtained from oat husks, as well as the effectiveness of our hydrothermal pretreatment to produce these products, a comprehensive analysis of their chemical composition is imperative. The corresponding data are presented in Table 1. Similar to many other lignocellulosic materials, the major constituents of oat husks are cellulose, hemicellulose (mostly xylan), and lignin. The major part consists of glucan, followed by xylan, with nearly half the quantity, while lignin is present in the smallest amount. With such a quantity of carbohydrates and a low lignin content, the oat husks are promising feedstock for the production of C6-derived bioproducts, such as bioethanol, 5-HMF, levoglucosanone, etc., as well as the production of C5-derived platform chemicals, such as furfural. Based on the obtained data, the theoretical amount of bioethanol, 5-HMF, and furfural that can be produced from one oven-dried ton of oat husks are 221.9, 304.6, and 197.0 kg, respectively. Placing a strategic emphasis on the selective transformation of oat husk carbohydrates into multiple bioproducts could yield significant benefits. By focusing on the selective and precise conversion processes, there is a substantial opportunity to achieve markedly improved economic returns. Simultaneously, this approach will contribute significantly to environmental sustainability by mitigating pollution.

In the literature, it can be found that the acid-insoluble lignin content in oat husks varies between 10.5 and 23.1% of o.d.m. Acid-soluble lignin is approximately 2.5% of o.d.m., and hemicellulose content in the literature varies between 24.0 and 35.1% of o.d.m. Compared with the literature, the oat husk sample that was used in this work is more enriched in terms of hemicelluloses and cellulose, making it a prospective feedstock for furfural production [36].

### 3.2. Furfural Production

Sulfuric acid is mostly used as a catalyst to obtain industrially produced furfural. The lignocellulosic waste containing sulfuric acid is utilized to generate water vapor for the furfural manufacturing process, resulting in SOx, which is hazardous to the environment [37]. As a result, in our experiments, we employed phosphoric acid as a catalyst, which is extensively used in food and other sectors (beverages, fertilizers, detergents, etc.). Similarly, the phosphoric acid that remains in the LC residue does not need to be neutralized; we have demonstrated that it does not interfere with enzymes when working with birch wood and conducting enzymatic hydrolysis [38].

To find the optimal catalyzed hydrothermal pretreatment process parameters to produce furfural from the oat husks, a central composite circumscribed (CCC) design and a response surface methodology were used. The effect of four factors (catalyst conc. (c), temperature (T), catalyst amount (m), and treatment time (τ)) on the C5 carbohydrates conversion into furfural was analyzed. The obtained furfural yield was expressed as the yield from the oven-dried mass (% of o.d.m.). The obtained yield of furfural according to the CCC design experimental runs is shown in Table 2.

The results of 26 experiments indicated that furfural was produced in the range of 0.55–11.84% o.d.m. A high furfural yield (10.67–11.67% o.d.m.) was achieved at a reaction temperature of 155–170 °C, a catalyst concentration of 25–35% *v*/*v*, a catalyst amount of 4–6 wt.%/wt. oat husk, and a reaction time of 30–50 min. Compared with the literature, data about obtaining a furfural amount from oat husks using 1% H_2_SO_4_ for 120 min at 180 °C is only 4.8% o.d.m. [39].

At a higher reaction temperature (185 °C) and reaction time (30 min), the furfural yield started to reduce. Therefore, it allows us to conclude that the optimal treatment parameters for producing furfural from oat husks are in the range of previously selected variable factors.

Without furfural, acetic acid, formic acid, levulinic acid, and 5-HMF were also identified in the obtained hydrolysate. Acetic acid was a major by-product sate (see Table 2). Meanwhile, formic acid, levulinic acid, and 5-HMF were minor by-products (see Figure 3). It is significant that at the highest yields of furfural (>10% o.d.m.), formic acid is formed more than other by-products. This can be explained by the fact that formic acid is a degradation product of furfural. Consequently, a higher amount of formic acid in the hydrolysate indicates that secondary reactions occur during furfural production. To prevent these secondary reactions, furfural removal from the reaction zone must be increased. This could be achieved with a higher steam flow rate. Such an approach would increase water consumption. Therefore, this opens further research opportunities about the potential benefits of higher steam flow.

To evaluate the influence of selected treatment parameters on furfural yield, the design matrix of experimental conditions with the corresponding furfural yields (Table 2) was subjected to regression analysis, generating the following quadratic equation in actual values:Furfural = -108.50964 + 2.13554(T) − 10.93677(m) − 0.075354(c) + 1.34504(T) + 0.568542(τ·m) + 0.052100(τ·c) − 0.032683 (τ·T) + 0.003062(m·c) + 0.021292(m·T) − 0.002492 (c·T) − 0.061504(T^2^) + 0.395781(m^2^) − 0.004819(c^2^) − 0.002292(T^2^) − 0.000921(τ·m·T) − 0.003887 (τ^2^·m) − 0.000874(τ^2^·c) + 0.000673(τ^2^·T) − 0.016094(τ·m^2^) (1)

According to the ANOVA test, the model presented a high value for the regression coefficient (R^2^ = 0.99). The value of the adjusted determination coefficient (adjusted R^2^ = 0.98) was also high, indicating a high significance of the model.

As indicated in Figure 4 and Figure 5, the furfural production yield increased by increasing all variables. As expected, the lowest amount of furfural was formed at the lowest values of variable factors. At the lowest catalyst amount and concentration (Figure 4A), the impact of treatment temperature and time increase was much more significant than it was at the highest catalyst concentration and amount studied. At medium conditions (Figure 4B), the reciprocal interaction of treatment time and temperature on the furfural yield was almost linear. In addition, under these conditions of catalyst concentration and amount, it can clearly be seen that the optimal treatment time and temperature should be searched at 158–170 °C and 35–40 min intervals. This is confirmed by the fact that if the amount and concentration of the catalyst are at higher studied values (Figure 4C), then the range of optimal parameters increases (25–40 min, 152–170 °C). On the other hand, the amount of furfural obtained in this treatment time and temperature range was lower than at the catalyst concentration of 25% and amount of 6 wt.%. Therefore, optimal conditions for catalyst concentration and amount to produce furfural with a higher yield must be sought in the ranges of 25–35% *v*/*v* and 6–8 wt.%.

The interaction between catalyst concentration and amount on the furfural yield at the lowest, middle, and highest treatment temperature and time according to the obtained mathematical model is illustrated in Figure 5. The obtained plot at the temperature of 140 °C and treatment time of 20 min (Figure 5A) shows that the surface is flat, suggesting that within the studied range of catalyst amounts and concentrations, there was almost no interaction effect on the furfural yield. The furfural yield remained constant at specific levels of catalyst amount and concentration, as indicated by the lack of slope on the surface. By increasing the treatment temperature and time up to medium values (Figure 5B), the interaction between the catalyst concentration and amount on the furfural yield indicates a slight upward slope and leads to a higher yield of furfural by increasing these two variables. This plot also indicates that the changes in catalyst concentration have a lower impact on the furfural yield than the catalyst amount in the reaction zone. The furfural amount can be increased from 6 to 8.5% o.d.m. by the increase in the catalyst concentration and its amount in the reaction zone.

At the highest studied temperature and treatment time, according to the obtained mathematical model, it shows a pronounced curvature, implying a strong interaction between catalyst amount and concentration. As shown by the peak in the surface, the highest yield of furfural (12.3% o.d.m.) is predicted at the catalyst amount of 6.2% o.d.m. and a catalyst concentration of 23%. Recalculating this predicted furfural amount on the theoretically possible amount, it can be said that such a treatment approach will allow us to convert 62% of available oat husk C5 carbohydrates into furfural in a relatively short time. 

In summary, temperature is the most significant factor influencing furfural yield, followed by treatment time, catalyst amount, and finally, catalyst concentration, which has the lowest influence. To determine the most suitable processing parameters for furfural production from oat husks, it is essential to evaluate the effect of the furfural removal rate from the reaction zone at narrower intervals for the previously mentioned treatment parameters. In our previous studies [32,33,34,35,40,41], we focused on the characterization of lignocellulosic (LC) biomass obtained after furfural production from birch wood chips. This study aimed to optimize the conditions for furfural production while also assessing the quality of the residual LC material for further valorization. Therefore, the main difference was the biomass used in the experiments. Both studies highlighted the significant impact of temperature and catalyst concentration on furfural yield. In the birch wood study, higher temperatures (175 °C) and specific catalyst concentrations (70%) were optimal for furfural production, which is consistent with the trend observed in the oat husks data, where temperature was the most influential factor. The treatment time and catalyst amount also play a crucial role in both studies. Longer treatment times and higher catalyst amounts were associated with higher furfural yields in both cases, although the optimal values differ due to the distinct nature of the feedstocks (oat husks vs. birch wood). Different types of biomasses have varying compositions, which can influence the efficiency of furfural production. Birch wood, which we studied previously, being a hardwood, has different acid-soluble lignin and acid-insoluble lignin content (19.42% and 3.71%, respectively) when compared to oat husks (2.99% and 12.99%, respectively), which can affect biomass processing efficiency, indicating the need for separate reaction parameter optimization. Glucan and xylan content in oat husks (39.17% and 23.58%) compared to birch wood chips (37.84% and 21.96%) is similar, indicating that the furfural yield after the optimization of process parameters should be similar for both types of feedstocks [32,35].

### 3.3. Chemical Composition of Lignocellulosic Residue

One of the targets of this study was to selectively remove C5 carbohydrates from oat husks during the pretreatment process and preserve glucan (cellulose) for further processing. To determine the efficiency of the treatment process from this point of view, the chemical composition was analyzed. The obtained data are collected in Table 3. 

The results indicate that 2–32% of oat husks’ initial mass after the pretreatment stage was removed. In severe treatment conditions, these losses are greater than in milder ones. These losses are related to the removal of volatile components, the conversion of acetyl groups to acetic acid, as well as the degradation of C5 and C6 carbohydrates during processing. Acid-insoluble residue, glucan, and xylan form the major part of oat husks lignocellulosic residue. 

Comparing the chemical composition of oat husks lignocellulosic residue (Table 3: Run 4, 5, 9, 13, 22, and 23), where furfural yield was higher than 10% o.d.m., almost all C5 carbohydrates were removed. Unfortunately, the glucan content was also irretrievably lost if converted to the original quantity. The acid-insoluble fraction has greatly increased due to concentration effects and the formation of pseudo-lignin due to a reaction between lignin and carbohydrates in acidic conditions, which could prevent access to the remaining glycan in the further stages of oat husk processing in some effect.

Although oat husk LC residue after furfural production has not been previously studied, when compared to other feedstocks, it can be seen that the obtained results show similar tendencies to those of other authors’ works. The composition of LC residue can be observed where most of the hemicellulose for the feedstock is consumed during the furfural production process, and the remaining part exists in the form of short-chain pentose (~2%). The rest of the LC residue mainly consists of lignin (~40%), cellulose (~35%), trace elements (2–5%), acetic acid (~4%), and catalyst residuals [42].

To better evaluate the effect of treatment process parameters on the losses of glucan (in the following figures expressed as cellulose), the design matrix of experimental conditions with the corresponding glucan yield in the lignocellulosic residue (Table 3) was subjected to regression analysis using Design Expert software, generating the following quadratic equation in actual values:Cellulose = +376.64249 − 2.87534(τ) − 8.36260(m) − 1.22988(c) − 4.34499(T) + 0.011256 (τ·c) + 0.018771(τ·T) + 0.059438 (m·T) + 0.006304 (c·T) + 0.012286 (T^2^) (2)

According to the ANOVA test, the model is simpler but less precise than in the case of furfural. The regression coefficient is 0.96, but the value of the adjusted determination coefficient is 0.94. Nevertheless, the necessary conclusions can be drawn.

As shown in Figure 6, the cellulose loss drastically increases along with the increasing severity of the reaction medium. The main factors that affect the losses of cellulose are treatment time, temperature, and the catalyst amount. As can be seen in Figure 6, cellulose degradation occurs almost linearly with an increase in treatment temperature and time. The smallest cellulose losses (1.6–17.3% from the initial amount) are achievable at the lowest catalyst concentration and amount (Figure 6A). 

Under the same conditions, the amount of furfural obtained is 4.2–52.9% of the theoretically possible amount. Therefore, if we look at the concept of oat husk biorefining, where furfural is obtained from C5 carbohydrates and cellulose is preserved for further processes, a compromise must be made. As shown in Figure 4B,C, increasing the furfural yield requires increasing the catalyst concentration and amount, which, in turn, increases cellulose losses to an even greater extent (Figure 6B,C).

## 4. Conclusions

The highest furfural yield (11.84% of o.d.m.) was observed under conditions of 15% catalyst concentration, 170 °C temperature, 8 wt.% catalyst amount, and 30 min treatment time, in these cases, acetic acid yields are 2.64% from o.d.m.The optimal glucan yield, while preserving a high furfural yield, is attained in experiment No. 4, where the yields of acetic acid, furfural, and glucan are 2.70%, 10.67% and 38.64%, respectively, at the technological pretreatment process parameters of 25% catalyst concentration, 6 wt.% catalyst amount, 155 °C temperature, and steam flow rate in the reaction zone 100 mL·min^−1^.Not all biomass feedstocks can be treated with the same pretreatment method in the context of biorefining. As a result, individual studies are needed for every raw material.For optimal furfural production from oat husks, the most important reaction parameters were maximal reaction temperature and time, which in turn leads to a higher loss of cellulose in LC residue. To determine the optimal balance between furfural yield and cellulose loss in LC residue, economic factors should be taken into account.

## Figures and Tables

**Figure 1 polymers-16-00707-f001:**
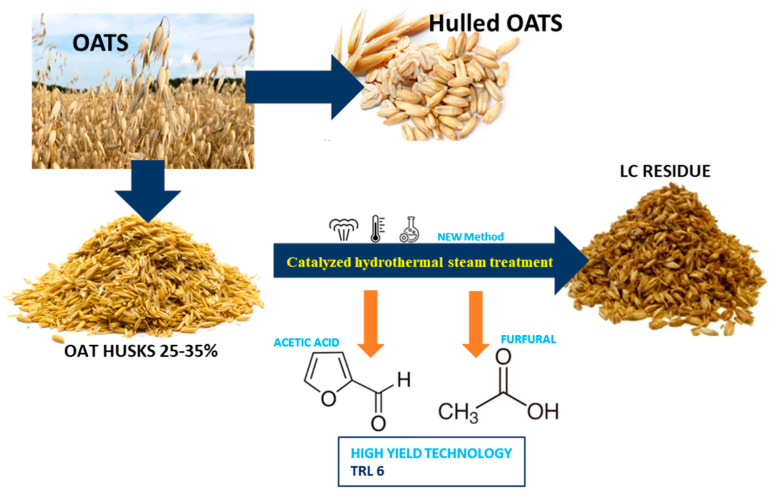
Scheme of the technological process.

**Figure 2 polymers-16-00707-f002:**
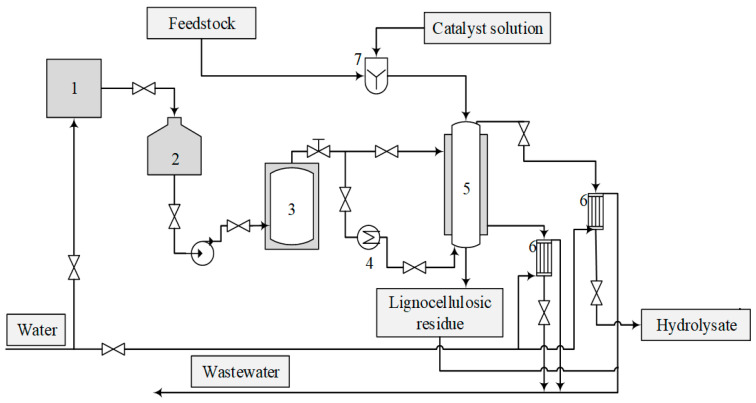
Scheme of the reactor system.

**Figure 3 polymers-16-00707-f003:**
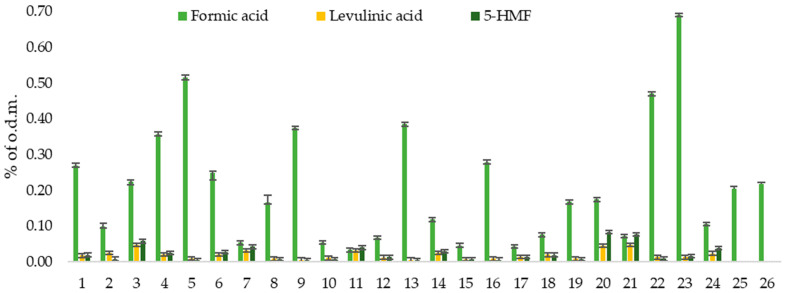
The admixture yield (% of o.d.m.) in the hydrolysis condensates.

**Figure 4 polymers-16-00707-f004:**
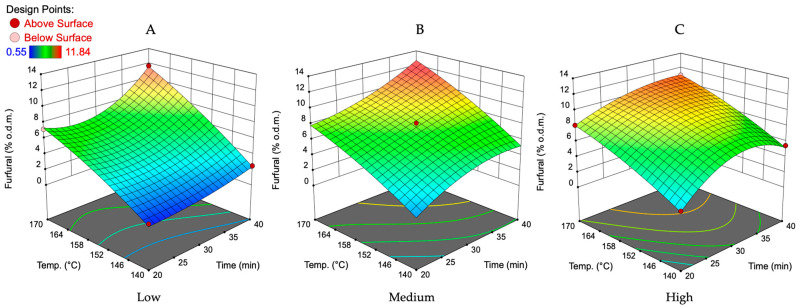
The effect of treatment time and temperature on the furfural yield at the lowest (**A**), medium (**B**), and highest (**C**) studied catalyst amount and catalyst concentration according to the obtained mathematical model. The lowest catalyst amount—4 wt.%, medium—6 wt.%, and highest—8 wt.%. The lowest catalyst concentration—15%, medium—25%, and highest—35%.

**Figure 5 polymers-16-00707-f005:**
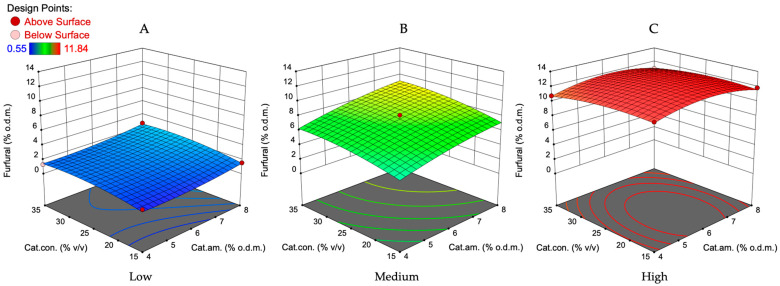
The effect of catalyst amount and catalyst concentration on the furfural yield at the lowest (**A**), medium (**B**) and highest (**C**) studied treatment time and temperature according to the obtained mathematical model. The lowest treatment time—20 min, medium—30 min and the highest—40 min. The lowest temperature—140 °C, medium—155 °C and highest—170 °C.

**Figure 6 polymers-16-00707-f006:**
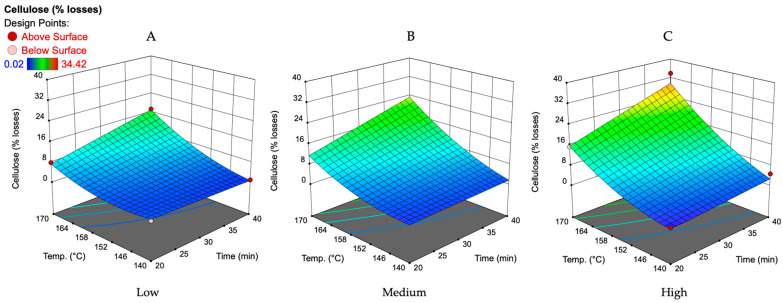
Effect of treatment time and temperature on the cellulose loss at the lowest (**A**), medium (**B**), and highest (**C**) studied catalyst amount and catalyst concentration according to the obtained mathematical model. The lowest catalyst amount—4 wt.%, medium—6 wt.%, and highest—8 wt.%. The lowest catalyst concentration—15%, medium—25%, and highest—35%.

**Table 1 polymers-16-00707-t001:** Chemical composition of oat husks.

Compound	Amount (% of o.d.m.)
Extractives (acetone)	3.27 ± 0.06
Glucan	37.44 ± 0.14
Xylan	23.58 ± 0.11
Galactan	0.55 ± 0.05
Arabinan	3.51 ± 0.02
Mannan	0.49 ± 0.01
Acid-insoluble residue (Klason lignin)	12.99 ± 0.06
Acid-soluble lignin	2.99 ± 0.04
Ash	4.90 ± 0.02
Acetyl groups	2.18 ± 0.03

**Table 2 polymers-16-00707-t002:** Furfural and acetic acid yield in the hydrolysis condensates according to the CCC experimental design.

Run	Catalyst Conc., (c)	Temperature, (T)	Catalyst Amount, (m)	Treatment Time, (τ)	Furfural	Acetic Acid
	%	°C	wt.%	min	% of o.d.m.	% of o.d.m.
1	25	155	10	30	8.70 ± 0.06	2.51 ± 0.01
2	35	140	4	40	3.14 ± 0.02	1.49 ± 0.01
3	25	155	6	30	8.15 ± 0.05	2.61 ± 0.02
4	25	155	6	50	10.67 ± 0.08	2.70 ± 0.03
5	25	185	6	30	10.14 ± 0.08	2.69 ± 0.04
6	45	155	6	30	7.33 ± 0.05	2.29 ± 0.03
7	25	125	6	30	0.55 ± 0.01	1.17 ± 0.01
8	15	170	4	20	7.20 ± 0.06	1.92 ± 0.02
9	15	170	4	40	11.67 ± 0.08	2.66 ± 0.04
10	25	155	6	10	2.30 ± 0.02	0.98 ± 0.02
11	15	140	4	20	0.92 ± 0.01	0.78 ± 0.03
12	5	155	6	30	3.63 ± 0.03	1.33 ± 0.01
13	15	170	8	30	11.84 ± 0.07	2.64 ± 0.01
14	25	155	2	30	3.33 ± 0.02	1.24 ± 0.03
15	15	140	8	20	1.54 ± 0.01	1.25 ± 0.01
16	35	170	8	20	8.12 ± 0.06	2.20 ± 0.02
17	35	140	4	20	1.30 ± 0.01	0.90 ± 0.02
18	35	140	8	20	2.64 ± 0.01	1.67 ± 0.01
19	35	140	8	40	5.50 ± 0.03	2.34 ± 0.03
20	15	170	8	20	8.72 ± 0.05	2.10 ± 0.02
21	15	140	4	40	2.50 ± 0.01	1.44 ± 0.01
22	35	170	4	40	10.78 ± 0.06	2.53 ± 0.03
23	35	170	8	40	10.96 ± 0.06	2.76 ± 0.02
24	15	140	8	40	4.22 ± 0.03	2.28 ± 0.02
25	25	155	6	30	8.06 ± 0.04	2.28 ± 0.03
26	35	170	4	20	6.99 ± 0.05	1.88 ± 0.02

**Table 3 polymers-16-00707-t003:** The chemical composition of obtained oat husks LC after furfural production.

Run	Oat Husks LC	Acid-Insoluble Residue	Glucan	Xylan	Arabinan	Galactan	Mannan	Acetyl Groups
	% o.d.m.	% LC						
1	77.64	31.56 ± 0.12	44.38 ± 0.12	14.14 ± 0.35	1.35 ± 0.09	1.42 ± 0.03	0.24 ± 0.05	0.37 ± 0.07
2	93.05	20.92 ± 0.15	38.64 ± 0.15	27.83 ± 0.15	1.74 ± 0.05	1.21 ± 0.02	0.12 ± 0.08	1.09 ± 0.01
3	82.85	28.02 ± 0.20	42.62 ± 0.22	18.18 ± 0.11	1.40 ± 0.17	1.13 ± 0.04	0.19 ± 0.00	0.50 ± 0.02
4	75.82	32.28 ± 0.21	43.61 ± 0.32	13.01 ± 0.45	0.81 ± 0.01	1.58 ± 0.16	0.07 ± 0.00	0.36 ± 0.01
5	68.04	49.30 ± 0.35	36.09 ± 0.40	1.50 ± 0.45	0.37 ± 0.03	1.68 ± 0.14	0.11 ± 0.03	0.25 ± 0.01
6	82.13	28.58 ± 0.34	42.97 ± 0.45	17.02 ± 0.47	1.21 ± 0.03	1.26 ± 0.13	0.10 ± 0.05	0.52 ± 0.03
7	97.37	17.39 ± 0.21	38.87 ± 0.40	31.60 ± 0.18	2.43 ± 0.05	1.43 ± 0.09	0.06 ± 0.01	1.31 ± 0.01
8	79.43	29.40 ± 0.15	43.43 ± 0.32	14.87 ± 0.16	1.06 ± 0.02	1.26 ± 0.08	0.13 ± 0.02	0.54 ± 0.03
9	70.69	37.21 ± 0.20	43.78 ± 0.50	7.71 ± 0.07	0.62 ± 0.01	1.24 ± 0.08	0.20 ± 0.03	0.32 ± 0.01
10	93.42	21.01 ± 0.15	38.53 ± 0.42	27.63 ± 0.36	1.83 ± 0.02	1.37 ± 0.17	0.03 ± 0.00	1.11 ± 0.03
11	96.66	17.66 ± 0.10	38.88 ± 0.30	29.69 ± 0.19	1.95 ± 0.05	1.39 ± 0.10	0.00 ± 0.00	1.38 ± 0.02
12	77.73	23.81 ± 0.20	44.34 ± 0.40	23.21 ± 0.23	1.58 ± 0.02	1.38 ± 0.04	0.11 ± 0.01	0.65 ± 0.02
13	68.37	41.84 ± 0.50	41.93 ± 0.50	5.80 ± 0.10	0.60 ± 0.07	1.12 ± 0.10	0.24 ± 0.04	0.27 ± 0.02
14	92.65	22.01 ± 0.35	39.72 ± 0.30	26.00 ± 0.14	1.48 ± 0.03	1.22 ± 0.04	0.08 ± 0.02	1.31 ± 0.01
15	91.82	18.78 ± 0.16	41.00 ± 0.40	29.11 ± 0.20	2.17 ± 0.02	1.48 ± 0.10	0.09 ± 0.01	0.81 ± 0.03
16	74.24	38.28 ± 0.25	42.68 ± 0.30	7.94 ± 0.15	0.76 ± 0.07	1.31 ± 0.03	0.16 ± 0.01	0.35 ± 0.02
17	96.94	19.02 ± 0.18	38.98 ± 0.35	29.35 ± 0.07	2.03 ± 0.03	1.15 ± 0.03	0.06 ± 0.01	1.50 ± 0.03
18	91.06	20.78 ± 0.20	41.02 ± 0.40	27.68 ± 0.01	1.98 ± 0.02	1.57 ± 0.09	0.12 ± 0.01	0.87 ± 0.02
19	85.52	24.07 ± 0.30	41.77 ± 0.46	22.75 ± 0.27	1.70 ± 0.05	1.29 ± 0.04	0.10 ± 0.03	0.50 ± 0.03
20	75.20	34.27 ± 0.36	44.27 ± 0.36	10.27 ± 0.08	1.01 ± 0.01	1.34 ± 0.03	0.19 ± 0.03	0.34 ± 0.01
21	92.34	20.04 ± 0.20	40.19 ± 0.45	27.73 ± 0.37	1.88 ± 0.02	1.37 ± 0.02	0.17 ± 0.02	1.07 ± 0.04
22	72.44	41.01 ± 0.40	40.73 ± 0.50	7.80 ± 0.33	0.54 ± 0.02	1.28 ± 0.00	0.28 ± 0.02	0.33 ± 0.02
23	68.13	48.24 ± 0.50	36.27 ± 0.30	5.00 ± 0.21	0.33 ± 0.01	1.33 ± 0.07	0.06 ± 0.02	0.24 ± 0.01
24	88.86	21.87 ± 0.22	41.82 ± 0.25	24.78 ± 0.20	1.82 ± 0.01	1.48 ± 0.13	0.12 ± 0.02	0.48 ± 0.04
25	82.74	27.74 ± 0.20	42.36 ± 0.48	18.27 ± 0.19	1.39 ± 0.02	1.28 ± 0.09	0.15 ± 0.02	0.48 ± 0.02
26	79.12	32.40 ± 0.30	43.02 ± 0.45	12.38 ± 0.22	0.82 ± 0.12	1.15 ± 0.06	0.23 ± 0.02	0.58 ± 0.01

## Data Availability

The raw data supporting the conclusions of this article will be made available by the authors on request.

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
