# Peer review of "Catalyzed Hydrothermal Pretreatment of Oat Husks for Integrated Production of Furfural and Lignocellulosic Residue"

_polymers, 2024, doi:10.3390/polym16050707_

Round 1

Reviewer 1 Report

Comments and Suggestions for Authors

This paper is well organised and prepared. There are only two portions should be improved.

1. The abstract was not clear on the method, the relevant information should be added.

2. The comparasion between orthophosphoric acid method and sulfuric acid method should be provided.

Author Response

Dear Reviewer (1),

Your comments about our manuscript were considered. Here is the response to your comments:

Question: The abstract was not clear on the method, the relevant information should be added.

Answer: Corrected in the text.

Question: The comparison between orthophosphoric acid method and sulfuric acid method should be provided.

Answer: We cannot provide such data because we used orthophosphoric acid for oat husks. Sulfuric acid is widely used in the industrial production of furfural, the residue of lignocellulose is burned, because the cellulose has decomposed in it by 40-50%. This creates SOx polution. It is also known that the industrial yield of furfural is 50-60% of the theoretical one, but we can reach 70-80%.

Reviewer 2 Report

Comments and Suggestions for Authors

The article polymers-2871698 is devoted to the conversion of oat husks into furfural and spent lignocellulosic residue.

The topic is relevant, and the authors have specialized in the production of furfural for many years. Oat husk is a traditional raw material for the production of furfural, so the development of new progressive methods for isolating it and obtaining several commercial products at the same time is not only scientific, but also practical.

The strengths of the work are the proven methodology, replicated on other types of raw materials. The authors casually compare oat husks with birch chips, line 329, but do not conclude whether the forecast for furfural yield from one type of raw material can be transferred to another type of raw material if the chemical compositions of these types of raw materials are similar? If yes, then the proposed method is universal and significant success has been achieved in lignocellulose processing. And if not, then it is necessary to conclude that it is impossible to transfer the yield forecast from one type of raw material to another and the research should be continued, which is also good.

I have a critical methodological remark:

1) Clause 2.4, clause 3.2, lines 281-295. It is not clear what the catalyst concentration and the amount of catalyst are and why they vary at the same time? Is the concentration of phosphoric acid expressed as a percentage by weight? Is the amount of phosphoric acid expressed as a percentage of the weight of the raw material? Concentration is a value that quantitatively characterizes the content of a component in a mixture. That is, the concentration and amount of catalyst can be brought into a single measurement system and these are not two parameters, but one.

Other notes:

2) The title of the manuscript should be changed and instead of the term “cellulose” the term “spent lignocellulosic residue” should be used, because this is the term the authors rightly use in the manuscript.

3) The article is stated in Topic “Green and Sustainable Chemistry”, in which case it should be justified why forphoric acid is greener than other catalysts.

4) Abstract now has nothing to do with the materials of the manuscript and looks like an introduction to a literature review. Reflect in the Abstract the novelty, research methodology and results obtained.

5) Lines 86-97 97. The authors set a goal to obtain furfural and acetic acid, but do not further draw conclusions on acetic acid. They do not provide correlation equations for the yield of acetic acid depending on the experimental conditions and the yield of acetic acid depending on the yield of furfural.

6) Information about the fate of lignin should also be added in the introduction. It is known that pseudolignin is formed under harsh conditions and the authors further write about this. What is the goal of the work? Apparently, the spent lignocellulosic residue should contain little pseudolignin if it is planned to obtain nanocellulose from it?

7) Lines 103-105. Is this your hypothesis? This should be said and attention should be paid to the achievement or not of this hypothesis. Also draw a conclusion in conclusion.

8) Clause 2.3. Please provide a link to the method for determining acid-soluble lignin. You provide data on acid-soluble lignin in Table 1.

9) Table 2. Please indicate which experiments were done in the center of the plan. You write about this in the methodology.

10) Point 2.3 Indicate how much oat husk you put in one cycle into the reactor chamber?

11) In paragraph 3.1, compare the chemical composition of oat husks with literature data.

12) In paragraph 3.2, compare the yield of furfural not with your data on other types of raw materials, but with the yield of furfural from oat husks obtained using other methods, with literature data.

13) Clause 3.3. Please compare the chemical composition of spent lignocellulosic residue with literature data.

14) The conclusions are not consistent with the task. The production of three products (furfural, acetic acid and cellulose) was claimed, but optimization was carried out to obtain one furfural. Either rewrite the goal or change the optimization parameter.

15) Now the work is trivial, the conditions for obtaining furfural have been optimized, but what is new?

Author Response

Dear Reviewer (2),

Your comments about our manuscript were considered. Here is the response to your comments:

Question: The strengths of the work are the proven methodology, replicated on other types of raw materials. The authors casually compare oat husks with birch chips, line 329, but do not conclude whether the forecast for furfural yield from one type of raw material can be transferred to another type of raw material if the chemical compositions of these types of raw materials are similar?

Answer: The parameters of the pretreatment process can be similar, but not identical, because not only the chemical composition of the raw material but also the structure of the raw material is different. It is also important and essential what the next processing step will be, whether it will be fiber or, for example, enzymatic hydrolysis. As we can see from the table below, the amount of xylan is different and in the case of oat husk a higher yield of furfural has been achieved.

Question: If yes, then the proposed method is universal and significant success has been achieved in lignocellulose processing. And if not, then it is necessary to conclude that it is impossible to transfer the yield forecast from one type of raw material to another and the research should be continued, which is also good.

Answer: You are right. There is no universal method. Thanks for the suggestion. We made corrections in the conclusions.

Question: Clause 2.4, clause 3.2, lines 281-295. It is not clear what the catalyst concentration and the amount of catalyst are and why they vary at the same time? Is the concentration of phosphoric acid expressed as a percentage by weight? Is the amount of phosphoric acid expressed as a percentage of the weight of the raw material? Concentration is a value that quantitatively characterizes the content of a component in a mixture. That is, the concentration and amount of catalyst can be brought into a single measurement system and these are not two parameters, but one.

Answer: We have discussed this issue a lot. In our case, these are two different parameters of the pretreatment process, one is the concentration of the relevant acid, while the other is how much of the relevant acid is used in the process from the absolutely dry mass. We will gladly listen to suggestions on how to combine these parameters into one. If, for example, 50% H3PO4 is used as catalyst in a volume of 3% of the mass of absolutely dry raw material. That’s means that we are using 50% H3PO4 in amount of 30g of H3PO4 per 1000g of raw material. Otherwise there are needed the ration g material per g of H3PO4

Question: The title of the manuscript should be changed and instead of the term “cellulose” the term “spent lignocellulosic residue” should be used, because this is the term the authors rightly use in the manuscript.

Answer: We have changed the title for the manuscript to the following “Catalyzed hydrothermal pretreatment of oat husks for integrated production of furfural and lignocellulosic residue”.

Question: The article is stated in Topic “Green and Sustainable Chemistry”, in which case it should be justified why phosphoric acid is greener than other catalysts.

Answer: Worldwide, sulfuric acid is used in the production of furfural. Which is harmful to the environment due to SOx. Therefore, we replace it with phosphoric acid, which is much more environmentally friendly, and phosphoric acid is used in many different food products.  Phosphoric acid is a food additive that acts as an acidity regulator and an acidulant. It can be found in soft drinks, dairy products, cereal bars, processed meats, cheese, and other processed foods. It should be noted that we plan to use the LC residue in the context of biorefining and it is important to preserve the cellulose in the LC residue. Our experience shows that using sulfuric acid as a catalyst leads to greater depolymerization and destruction of cellulose.

Question: Abstract now has nothing to do with the materials of the manuscript and looks like an introduction to a literature review. Reflect in the Abstract the novelty, research methodology and results obtained.

Answer: The abstract was changed.

Question: Lines 86-97 97. The authors set a goal to obtain furfural and acetic acid, but do not further draw conclusions on acetic acid. They do not provide correlation equations for the yield of acetic acid depending on the experimental conditions and the yield of acetic acid depending on the yield of furfural.

Answer: In this process, deacetylation occurs independently of the furfural formation process. These are different mechanisms. Acetic acid is like a by-product formed from biomass in the pretreatment process. The acetic acid yields are reported in Table 2.

Question: Information about the fate of lignin should also be added in the introduction. It is known that pseudolignin is formed under harsh conditions and the authors further write about this. What is the goal of the work?

Answer: Our pretreatment process is unique in that we are able to obtain a higher yield of furfural and at the same time reduce cellulose destruction 2-10% from initial cellulose in the lignocellulosic residue. This is also our goal, to achieve a high yield of furfural and preserve the cellulose in the lignocellulosic residue for further research.

Question: Apparently, the spent lignocellulosic residue should contain little pseudolignin if it is planned to obtain nanocellulose from it?

Answer: Yes, you are right, the lignocellulosic residue not only contains a high yield of cellulose but also contains pseudo-lignin. We planned to use the lignocellulose residue itself in the obtaining of nanocellulose. We tried that too, but more in-depth research is needed. We carried out these studies within the framework of a correspondingly small funding. We are planning to continue both studies on nanocellulose and 5-HMF research from lignocellulose residue.

Question: Lines 103-105. Is this your hypothesis? This should be said and attention should be paid to the achievement or not of this hypothesis. Also draw a conclusion in conclusion.

Answer: Yes, this is our hypothesis, which is based on the developed theory and mechanism of furfural formation and extraction in the pretreatment process, which was developed at the Latvian State Institute of Wood Chemistry, Polysaccharide department under the leadership of our professor N. Vedernikov. We continue the research of this hypothesis on many different raw materials and different catalysts for future integration in the context of biorefinery.

Question: Clause 2.3. Please provide a link to the method for determining acid-soluble lignin. You provide data on acid-soluble lignin in Table 1.

Answer: Information was added in the text.

Question: Please indicate which experiments were done in the center of the plan. You write about this in the methodology.

Answer: We added a reference in the text and highlighted the center of the plan in the table.

Question: Point 2.3 Indicate how much oat husk you put in one cycle into the reactor chamber?

Answer: In each experiment, we used the same amount of oat husks as 1360 g o.d.m. plus moisture and catalyst whose amount varied. 

Question: In paragraph 3.1, compare the chemical composition of oat husks with literature data.

Answer: Information added in the manuscript.

Question: In paragraph 3.2, compare the yield of furfural not with your data on other types of raw materials, but with the yield of furfural from oat husks obtained using other methods, with literature data.

Answer: We cannot compare with results obtained in laboratories. We can only compare with the results obtained in the pilot equipment, then it will be correct. In order to compare our method with other methods, they must be at the level of pilot plants not in lab scale.

Question: Clause 3.3. Please compare the chemical composition of spent lignocellulosic residue with literature data.

Answer: Oat husk has not been previously used as a feedstock for the simultaneous production of furfural and acetic acid. Therefore there is no available literature on this specific lignocellulose residue. It would be unwise to compare LC residues from different treatments and feedstocks.

Question: The conclusions are not consistent with the task. The production of three products (furfural, acetic acid and cellulose) was claimed, but optimization was carried out to obtain one furfural. Either rewrite the goal or change the optimization parameter.

Answer: The conclusions were changed in the manuscript.

Question: Now the work is trivial, the conditions for obtaining furfural have been optimized, but what is new?

Answer: Since we conduct studies in an original pilot plant rather than in laboratory reactors, the technology we developed has no competitors. Research is conducted using actual biomass rather than pure xylose, as numerous papers have noted. Thanks to our approach, we can now achieve a 20–25% increase in furfural's industrial yield. All currently used industrial processes break down 30–50% of the cellulose in the lignocellulosic residue during the furfural extraction process; this leaves the residue with no commercial value other than as fuel. Therefore, its application in the context of biorefining is limited. In fact, this is feasible with the technology we have created, as demonstrated on a pilot scale. Although oat husks as a raw material are not new, applying our method in this way is novel and creates a wealth of opportunities for the subsequent processing of biomass following furfural production.

Round 2

Reviewer 2 Report

Comments and Suggestions for Authors

The authors have made some changes. But I was surprised that after giving detailed professional answers to the reviewer, the authors did not change the text of the manuscript. The authors have interesting work, I want it to be read, noticed and quoted.

Therefore, I provide new comments that will help improve the quality of the manuscript.

1) The abstract should reflect unique information about the authors’ work, and not a statement of why the work is relevant. The long prelude about relevance in the abstract should be shortened. In their responses to the reviewer, the authors indicate that their study differs from the rest of the world in that the authors’ research was conducted in a unique setup under pilot conditions. This should definitely be reflected in the annotation.

2) Based on the annotation, phosphoric acid has advantages over sulfuric acid. In the discussion of the results, this should be justified, the results obtained on the use of phosphoric acid should be compared with the early results of the use of sulfuric acid.

3) The phrase from the annotation “As a result of the research, it was also possible to 23 reduce the destruction of cellulose to 11.69% of the output cellulose of oat husks” is not disclosed inside the manuscript. Was it possible to reduce the destruction of cellulose compared to what?

4) Information about phosphoric acid should be added to the production part. So that not only the reviewer, but also the readers know about this.

5) There is a big problem in your study: lignin after pretreatment becomes three times more than it was in the raw material. This should definitely be discussed. Such a large amount of pseudolignin can inhibit enzymatic hydrolysis very well.

6) It is imperative to compare the chemical composition of the lignocellulosic residue with literature data. It is clear that preprocessing has not been carried out under such conditions before. At a minimum, compare with sulfuric acid.

7) I will not find fault with the hypothesis, since it is included in the materials and methods, it is not necessary to prove it within the framework of this manuscript.

8) Indicate the experiments in the center of the plan as comments on the manuscript; it is obvious that the yellow color will be removed from the table.

9) Enter information about how much oat husk is placed into the chamber in one cycle in the text of the manuscript.

10) You should compare your furfural yield to other labs' yields from other feedstocks. At the same time, you can focus on the advantages of your research, since it was carried out in pilot conditions.

11) Conclusion 5 is not clear. It follows from this that you can do everything as with wood chips. Then why were the studies carried out?

Author Response

Dear Reviewer (2),

Your comments about our manuscript were considered (the main changes in the publication are marked in green). Here is the response to your comments:

1) Question: The abstract should reflect unique information about the authors’ work, and not a statement of why the work is relevant. The long prelude about relevance in the abstract should be shortened. In their responses to the reviewer, the authors indicate that their study differs from the rest of the world in that the authors’ research was conducted in a unique setup under pilot conditions. This should definitely be reflected in the annotation.

Answer: The abstract is changed in the text.

2) Question: Based on the annotation, phosphoric acid has advantages over sulfuric acid. In the discussion of the results, this should be justified, the results obtained on the use of phosphoric acid should be compared with the early results of the use of sulfuric acid.

Answer: Additional information added in the manuscript (lines 218 – 225 and 236 – 238).

3) Question: The phrase from the annotation “As a result of the research, it was also possible to 23 reduce the destruction of cellulose to 11.69% of the output cellulose of oat husks” is not disclosed inside the manuscript. Was it possible to reduce the destruction of cellulose compared to what?

Answer: Additional information added in the abstract.

4) Question: Information about phosphoric acid should be added to the production part. So that not only the reviewer, but also the readers know about this.

Answer: Information added in the manuscript. Lines 218 – 225.

5) Question: There is a big problem in your study: lignin after pretreatment becomes three times more than it was in the raw material. This should definitely be discussed. Such a large amount of pseudolignin can inhibit enzymatic hydrolysis very well.

Answer: Increase in lignin content can be explained by concentration of lignin due to removal of hemicellulose component. Also due to formation of pseudolignin in acidic conditions between carbohydrates and lignin. Although this can definitely impact enzymatic hydrolysis of obtained lignocellulose residue, it should not impact production of 5-HMF from this residue. Information added in the manuscript. Lines 361 – 364.

6) Question: It is imperative to compare the chemical composition of the lignocellulosic residue with literature data. It is clear that preprocessing has not been carried out under such conditions before. At a minimum, compare with sulfuric acid.

Answer: Information added in the manuscript. Lines 365-371.

7) I will not find fault with the hypothesis, since it is included in the materials and methods, it is not necessary to prove it within the framework of this manuscript.

Answer:  Main purpose of this manuscript has been given at the end of introduction section and reworked.

8) Question: Indicate the experiments in the center of the plan as comments on the manuscript; it is obvious that the yellow color will be removed from the table.

Answer: A change has been made. Lines 179-180.

9) Question: Enter information about how much oat husk is placed into the chamber in one cycle in the text of the manuscript.

Answer: It is now attached in the text, lines 142-143.

10) You should compare your furfural yield to other labs' yields from other feedstocks. At the same time, you can focus on the advantages of your research, since it was carried out in pilot conditions.

Answer:  Furfural yields from different feedstocks given in lines 54-64

11) Conclusion 5 is not clear. It follows from this that you can do everything as with wood chips. Then why were the studies carried out?

Answer: Authors concluded that conclusion 5 is not needed in the manuscript.